# Cutting Force Transition Model Considering the Influence of Tool System by Using Standard Test Table

**DOI:** 10.3390/s21041340

**Published:** 2021-02-13

**Authors:** Xi Chen, Dinghua Zhang, Qi Wang

**Affiliations:** The Key Laboratory of High Performance Manufacturing for Aero Engine, Northwestern Polytechnical University, Ministry of Industry and Information Technology, Xi’an 710072, China; chenxi15@mail.nwpu.edu.cn (X.C.); dhzhang@nwpu.edu.cn (D.Z.)

**Keywords:** milling, standard test table, cutting force transition, data application

## Abstract

The cutting force prediction model usually uses the classical oblique transformation method, which introduces the orthogonal cutting parameters into the oblique milling edge shape, and combines the geometric parameters of the tool to convert the orthogonal cutting force into the actual cutting force, thereby predicting the cutting force. However, this cutting force prediction method ignores the impact of tool vibration in actual machining, resulting in a large difference between the prediction model and the actual measurement. This paper proposes a cutting force conversion model considering the influence of the tool system. The proposed model fully considers the impact of tool vibration on the cutting force. On the basis of the orthogonal model, superimposing the additional cutting force generated by tool vibration makes the predicted value of the model closer to the actual cutting force. The results of milling experiments show that the conversion model can obtain higher prediction accuracy. Moreover, compared with the original conversion model, the accuracy of the proposed model is significantly improved.

## 1. Introduction

Machining is an important method in producing aeroengine parts, which is widely used in aerospace components processing [1]. The design and optimization of all aspects of the cutting process [2] are closely related to production quality and are the key to improve efficiency and reduce costs. Therefore, monitoring and analysis of machining process parameters are needed to avoid tool wear and improve machining quality. Research on machining process information usually studies cutting forces [3] or system dynamics [4].

The cutting force generated by the machining process of the machine tool is related to the machining state. Cutting force signal is a physical quantity commonly used in machining, which is used to evaluate the machining process and quality [5,6,7]. It is very important to obtain accurate cutting force to research chatter, overload, and tool wear [8]. Taner et al. [9] proposed a widely used cutting force model which can predict cutting force by identifying cutting force coefficients. Zhu et al. [10,11] proposed a method to analyze tool positioning changes and tool runout by calculating the undeformed cutting thickness and improved the model in subsequent research. Sun et al. [12] proposed a dynamic milling parameter prediction model considering the influence of deformation. The thin-walled parts and cutting tools were discretized into micro elements, and their dynamic effects were analyzed in each contact area. Based on the instantaneous cutting force model of tool contact point and the stiffness of machining system, Sahoo et al. [13] studied the influence of tool meshing area on cutting force. According to the tool path and the corresponding cutting force, Zhang et al. [14] proposed an efficient calibration procedure for cutter runout parameters and specific cutting force coefficients, which was an accurate prediction method for cutting force in 5-axis flank milling of a sculptured surface. Li et al. [15] proposed a dynamic model considering the cutting insert engagement based on the geometric characteristics of the workpieces and the tool path, which can overcome the poor accuracy of chatter prediction as well as the waste of processing efficiency. Wang et al. [16] and Zhu et al. [17] solved the differential equation by using the measured modal parameters to obtain the tool vibration information or vibration information per tooth during the machining process, which was superimposed on the feed rate to predict the cutting force. However, these cutting force models are used to predict the cutting force under specific processing conditions through experiments. When the processing conditions change, a lot of experiments are needed to calibrate the model coefficients, which makes the model coefficients unable to use directly. The main reason for this phenomenon is the influence of tool system modal information, measuring equipment, and the machine tool processing system.

The research on tool system dynamics [18] and machine tool performance [19] has received more attention. The performance of the machine tool is very important to the improvement of machining quality and efficiency [20]. The finite element method is used to establish a model close to the real machine tool structure, analyze its performance, and achieve the purpose of improving efficiency. In order to analyze more accurately, Zhang et al. [21] proposed the substructure response analysis method. The machine tool as a whole is regarded as the coupling of single parts, thus the dynamic information of the tool tip can be found. Ji et al. [22] proposed a tool tip dynamic characteristic prediction method, which considered the contact dynamic characteristics between the tool and the tool holder, using the existing tool hammer test and finite element model to predict the dynamic characteristics of the new tool tip. This method can not only predict the performance of the machine tool, but also be more convenient for the performance analysis of the machine tool with the tool spindle combination changed. Chen et al. [23] and Liu et al. [24] combined substructure coupling with deep learning and used the transfer learning method to predict tool tip dynamic information at different positions. The existing modal [25] analysis methods are limited to the static conditions of machine tools, and most of them are applied to the transition of machining state, and few of them are used to analyze the influence of cutting force. The working modal analysis is adapted to characterize the dynamic characteristics of the machine tool under working conditions. Therefore, it is necessary to study the dynamic characteristics of machine tools for machining process.

In the machining of thin-walled components with difficult to machine materials, the actual cutting conditions are closely related to materials, structures, processing, and processes [26,27]. At present, the problem of obtaining cutting parameters is that the data acquisition conditions [28,29] are different, which cannot be directly applied and compared in the heterogeneous process system of material structure process. Therefore, it is necessary to solve the unity, comparability, and generality of basic cutting data acquisition. However, the current basic cutting experiment data acquisition is mainly for limited specific parameters [30], such as cutting force, lack of sufficient physical data. In addition, various test equipment [31,32,33] and external environment interference during the test process affect the observation test results, which results in inaccurate data sources, large errors and low reliability of simulation results in the physical simulation of the cutting process.

This paper proposed a cutting force transition model considering the influence of a tool system. The standard cutting force obtained by the standard test table can be converted into the milling force in the actual milling process. Firstly, obtain the standard cutting force under the self-developed standard test table. Secondly, the dynamic parameters of the milling tool are tested on the machine tool through the hammer test, and the corresponding acceleration signal during the cutting process is measured. Finally, the standard cutting force is combined with the tool dynamic parameters, and the cutting force in the actual milling process is predicted through the proposed model. The experimental results show that the results are basically consistent, indicating the reliability of the method.

## 2. Transition Model of Standard Cutting Force under Standard Test Table

In the steady state cutting process, as the material is removed, the force acting on the workpiece can be divided into tangential force $F_{t}$ and radial force $F_{f}$. The forces under different cutting parameters can be expressed as follows:(1)$$\left\{\begin{array}{c} F_{t}=K_{tc}bh+K_{te}b\\ F_{f}=K_{fc}bh+K_{fe}b \end{array}\right.$$
where,$\;K_{tc}$ and $K_{te}$ are the tangential cutting force coefficients,$\;K_{fc}$ and $K_{fe}$ are the feed cutting coefficients, b is the cutting width, h is the cutting depth.

When the relationship between orthogonal cutting and milling is established, the cutting force coefficients are considered to be constant, and it can be regarded as only related to the material characteristics of the workpiece. When it is applied to the milling force model, the cutting force of milling can be obtained.

The cutting force of workpiece in milling can be divided into three directions: tangential force $F_{t}$, radial force $F_{r}$, and axial force $F_{a}$, as shown in Figure 1, these three forces could be expressed as follows:(2)$$\left\{\begin{array}{c} dF_{i,j,t}\left(\phi \right)=\left[K_{tc}h\left(\phi \right)+K_{te}\right]dz\\ dF_{i,j,r}\left(\phi \right)=\left[K_{rc}h\left(\phi \right)+K_{re}\right]dz\\ dF_{i,j,a}\left(\phi \right)=\left[K_{ac}h\left(\phi \right)+K_{ae}\right]dz \end{array}\right.$$
where, ϕ is the instantaneous rotatory angle of cutter. $h\left(\phi \right)$ is the instantaneous undeformed chip thickness on the jth disk of ith tooth. It can be expressed as follows:(3)$$h\left(\phi \right)=f_{z}\sin\left(\phi \right)$$

Transform cutting forces into static coordinate system by following equation:(4)$$\left[\begin{array}{c} dF_{i,j,x}\left(\phi \right)\\ dF_{i,j,y}\left(\phi \right)\\ dF_{i,j,z}\left(\phi \right) \end{array}\right]=\left[\begin{array}{ccc} -\cos\phi & -\sin\phi & 0\\ \sin\phi & -\cos\phi & 0\\ 0 & 0 & 1 \end{array}\right]\left[\begin{array}{c} dF_{i,j,t}\left(\phi \right)\\ dF_{i,j,r}\left(\phi \right)\\ dF_{i,j,a}\left(\phi \right) \end{array}\right]$$
where, $dF_{i,j,x}\left(\phi \right)$ is differential feed force,$\;dF_{i,j,y}\left(\phi \right)$ is differential normal force. $dF_{i,j,z}\left(\phi \right)$ is differential axial force. After a period of time, the sum of the forces acting on the tool can be expressed as follows:(5)$$\left[\begin{array}{c} F_{x}\left(\phi \right)\\ F_{y}\left(\phi \right)\\ F_{z}\left(\phi \right) \end{array}\right]=\overset{N}{\underset{i\;=\;1}{\sum }}\overset{M}{\underset{j\;=\;1}{\sum }}g\left(\theta \right)\left[\begin{array}{c} dF_{i,j,x}\left(\phi \right)\\ dF_{i,j,y}\left(\phi \right)\\ dF_{i,j,z}\left(\phi \right) \end{array}\right]$$
with:(6)$$g\left(\theta \right)=\left\{\begin{array}{c} 1,\;if\phi_{st}<\theta <\phi_{ex},\\ 0,\;otherwise.\; \end{array}\right.$$
where, $\phi_{st}$ is the entry angle and $\phi_{ex}$ is the exit angle.

## 3. Cutting Force Transition Model Considering Tool System Influence

As mentioned above, the cutting force coefficients identified on the standard test table can be considered to be obtained by the steady state cutting process. In steady state cutting, these parameters, such as cutting parameters (v, f, $a_{p}$), cutting layer parameters, tool angle are considered to be basically unchanged, but when the cutting process occurs impact and vibration, the above parameters change greatly with time. Under the cutting chatter condition, the tool workpiece angle, chip cross-section, tool and workpiece contact friction state, shear angle are all changed at any time, which caused the change of cutting force.

In the dynamic cutting process, due to factors such as tool vibration and workpiece material unevenness, the cutting force is not constant, but changes with time. Therefore, the cutting force can be divided into steady cutting force $F_{m}$ and dynamic cutting force changes with time$\;dF\left(t\right)$, that is,
(7)$$F\left(t\right)=F_{m}+dF\left(t\right)$$

During the dynamic cutting process, relative vibration occurs between the tool and the workpiece. The vibration can be divided into two components: one is that the component with the same direction of cutting speed changes the cutting speed between v+dv and v−dv, and changes the relative friction between the flank and the workpiece; the other is the component with the same direction as the feed speed causes the chip thickness to change in $h_{c}+dh_{c}$ and $h_{c}-dh_{c}$, resulting in a feed rate of $v_{f}+dv_{f}$ and $v_{f}-dv_{f}$. Each of the above changes can make the cutting force change in different degrees. No matter whether only considering one of them, or also considering some changes; meanwhile, the cutting force is always in$\;F_{m}+dF$ and $F_{m}-dF$. Therefore, the dynamic cutting force can be expressed as follows:(8)$$dF=K_{1}dh_{c}+K_{2}dv_{f}+K_{3}dv$$
where, $K_{1}={\left.\frac{\partial \mathrm{dF}}{\partial dh_{c}}\right|}_{dv_{f}=\mathrm{dv}=0}$,$\;K_{2}={\left.\frac{\partial \mathrm{dF}}{\partial dv_{f}}\right|}_{dh_{c}=\mathrm{dv}=0}$,$\;K_{3}={\left.\frac{\partial \mathrm{dF}}{\partial \mathrm{dv}}\right|}_{dh_{c}=dv_{f}=0}$,$\;K_{1}$ is the influence coefficient of chip thickness, $K_{2}$ is the influence coefficient of feed speed, $K_{3}$ is the influence coefficient of cutting speed.

In dynamic cutting, the variation of cutting speed depends on the vibration direction. If the vibration only occurs in the cutting direction, the vibration displacement will not change the cutting speed, but only change the chip thickness and feed rate. However, it is difficult to obtain the change of feed rate in the actual machining process, the dynamic cutting force can be simplified as only considering the change of chip thickness. Due to the stiffness of the spindle tool system, vibration will occur between the tool and workpiece, which will cause the change of chip thickness. Theoretically, the machine tool has vibration in three directions. The machine used in the experiment has stable axial stiffness, and the axial vibration can be ignored, see Figure 2. Base on this, the system can be simplified described as follows:(9)$$Ma_{\Delta }\left(t\right)+Cv_{\Delta }\left(t\right)+KX_{\Delta }\left(t\right)=F_{\Delta }\left(t\right)$$
where, $M=\left[\begin{array}{cc} m_{x} & 0\\ 0 & m_{y} \end{array}\right]$, $\;C=\left[\begin{array}{cc} c_{x} & 0\\ 0 & c_{y} \end{array}\right]$, $\;K=\left[\begin{array}{cc} k_{x} & 0\\ 0 & k_{y} \end{array}\right]$, $\;a_{\Delta }\left(t\right)=\left[\begin{array}{c} a_{\Delta x}\left(t\right)\\ a_{\Delta y}\left(t\right) \end{array}\right]$, $v_{\Delta }\left(t\right)=\left[\begin{array}{c} v_{\Delta x}\left(t\right)\\ v_{\Delta y}\left(t\right) \end{array}\right]$, $\;X_{\Delta }\left(t\right)=\left[\begin{array}{c} x_{\Delta }\left(t\right)\\ y_{\Delta }\left(t\right) \end{array}\right]$, $F_{\Delta }\left(t\right)=\left[\begin{array}{c} F_{\Delta x}\left(t\right)\\ F_{\Delta y}\left(t\right) \end{array}\right]$.

Tool vibration bringing variation of the undeformed cutting thickness that directly influences the dynamic fluctuation of cutting forces. To take this influence into account in the calculation of cutting forces, an additional displacement could be added to the ideal cutting thickness:(10)$$x_{\Delta }\left(t\right)=x\left(t\right)-h\left(t\right)$$
where,$\;x\left(t\right)$ is obtained by acceleration signals. $h\left(t\right)$ is the ideal cutting thickness.

The cutting force obtained by cutting force coefficients can be regarded as the standard cutting force, and the predicted cutting force can be obtained by adding the influence of the tool system, which can be described as follows:(11)$$F_{standard}\left(t\right)+F_{\Delta }\left(t\right)=F_{predicted}\left(t\right)$$

The flowchart of the proposed model calculation method is shown in Figure 3. The specific steps are as follows: (1) Carry out orthogonal cutting test on the self-developed standard test table to collect cutting force data as the basis of data conversion; (2) When converting from orthogonal cutting to milling, not only should the cutting force be mathematically integrated in the feed direction and axial direction, but also the influence of the tool system should be considered, and the tool dynamic parameters can be obtained by hammering test; (3) Using acceleration sensor to obtain tool displacement signal, combined with the dynamic equation, the predicted cutting force can be obtained by adding standard cutting force; (4) The accuracy of the cutting force conversion model is verified by comparing the measured force with the predicted force by experiments.

## 4. Experimental Verification and Discussion

### 4.1. Data Collection of Cutting Force and Identification of Cutting Force Coefficients

The orthogonal cutting experiment was carried out on the self-developed standard test table to obtain the standard cutting force data. The standard test table mainly consists of a tool and workpiece clamping module, motion control module, force acquisition module, and other parts. As shown in Figure 4, the standard test table can easily simulate orthogonal cutting. Its main principle is to convert the rotation of the motor into lateral movement. The workpiece holder on the moving table moves horizontally on the guide rail and fix the workpiece on the guide rail. The tool on the cross table produces relative movement, thus forming an orthogonal cutting process. The cutting thickness is adjusted by the inclined cross table at the bottom, and the cutting speed *v* is controlled by changing the pulse number of the servo motor per second. Kistler 9347C is a triaxial force transducer which is under the tool. The workpiece material is aluminum alloy 7475, which is the most widely used nonferrous metal material in modern aerospace parts [34,35]. Aluminum alloy 7475 with low-density and high-strength is the first material for lightweight parts and it is widely used in civil aircraft due to its good processability, good electrical and thermal conductivity. The cutting condition is dry cutting (see Table 1).

In order to obtain accurate cutting force coefficients, the experiment uses four groups of cutting parameters, which carried out 10 experiments under the same cutting conditions. The cutting force data of the standard test table is shown in Figure 5, and the identified cutting force coefficients are shown in Table 2.

The milling experiment was performed on the YH850 3-axis machining center to obtain the cutting force data during the milling process, the force signal is measured by Kistler 9257B, which is a multi-component dynamometer. The workpiece material is aluminum alloy 7475. The tool is SGO S550, a 3–blade end mill. It is made of a new type of ultra-fine tungsten steel base metal with high wear resistance and strength. It is specially used for high hardness and high speed cutting applications. For the failure of sensors in Z direction, only data in X and Y axis will be processed. The cutting condition is dry cutting. The cutter geometry and cutting parameters are shown in Table 3 and Table 4. The experiment platform construction is shown in Figure 6.

### 4.2. Comparison and Verification

The standard cutting force is calculated by using the cutting force coefficients, which are obtained from the standard test table (Table 2). Through the transfer model (Equations (1)–(5)), the milling force under different parameters can be calculated. Several typical results of standard and measurement are compared (Figure 7). Although the cutting conditions and machine tools have changed, the cutting force measured during the milling process (called the measuring force) and the cutting force transferred from the standard cutting force (called the standard force) are still relatively close. However, there are many differences between the standard force and the measured force, especially in the peak and valley positions of cutting force. The reason for this phenomenon is that the vibration of the tool causes the cutting thickness in the actual machining to change, which in turn causes the cutting force to change.

In addition, it is found in Figure 7 that the measured force is close to the standard force at the cutting in and cutting out stages of each tooth, and there is a huge difference between them at the middle cutting stage of each tooth. For example, Figure 7a shows the cutting forces in the second group of experiments. At the initial cutting in of the tool, the standard value of the cutting force is close to the measured single. When the chip thickness of each tooth reaches the maximum, the standard value and the measured value are quite different. The rule of variation is also applicable to Figure 7b,c. Therefore, the influence of the tool system cannot be ignored in the conversion of cutting force.

In order to identify the difference of the cutting force, the influence of the dynamic parameters of the tool is considered, which can be obtained by hammering test, see Table 5.

This paper uses the orthogonal cutting for identification to eliminate the influence of external factors on the cutting force coefficients. Therefore, in the process of tool rotary cutting, the measured cutting force always adds the influence of tool dynamic parameters. As the cutting thickness changes, the impact is constantly changing. Under the cutting parameters in Table 3, the cutting force and tool displacement during machining are collected. Figure 8 shows a section of tool displacement signal.

The dynamic parameters and displacement signals of the tool are substituted into Equations (8) and (10). After the influence of the tool system is superimposed, the measured cutting force is compared with the predicted cutting force calculated by proposed method. The results are shown in Figure 9. Considering the inhomogeneity of material and the contingency of measurement, the predicted value is in good agreement with the measured value. Obviously, the predicted value is distributed around the measurement, which shows that the model is reasonable and correct conclusions can be obtained. In Figure 9, the peak and valley of cutting force in X direction are similar in values, and the same in Y direction.

By comparing the standard value, measured value and predicted value of cutting force, the effectiveness of the proposed model can be seen. As shown in Figure 10, the yellow surface represents the standard cutting force under different cutting parameters, and the dark blue surface is at the bottom, which indicates the measured cutting force. The difference between the two surfaces is large, indicating that the standard cutting data has not been considered tool system impact in the actual machining. The light blue surface in the middle is the predicted value of the model proposed in this paper, and it is found to be closer to the measured value. The results show that the method of superimposing too system influence on standard cutting force is correct and feasible.

The detailed analysis is as follows: the errors calculated by Equations (12) and (13) are shown in Figure 11.
(12)$$error_{before}=\left|\frac{F_{standard}-F_{measured}}{F_{measured}}\right|$$
(13)$$error_{after}=\left|\frac{F_{predicted}-F_{measured}}{F_{measured}}\right|$$

The error between the standard and the measured value is obviously higher than the error between the predicted and the measured value through Figure 11, which indicates that the accuracy of the transition model is improved after the influence of the tool system is superimposed. The applicability of the proposed model is validated in a series of experiments with different cutting parameters.

The average value and deviation are shown in Table 6. The data in Table 6 also show that the conversion error of cutting force using proposed model is close to the measured cutting force. It can be found that, when ignoring the influence of the tool system, the cutting force conversion error is large, even exceeding 30% (Test No. 1, 5 etc.). The cutting force conversion error of the method proposed in this paper is significantly reduced, and the error is about 10%.

Since the tool system influence is the measured acceleration signal, the change is not regular, so the actual cutting thickness of each tooth is different in the tool rotary cutting. In order to further verify the accuracy of the model, the error of cutting force of each tooth under different cutting parameters should be analyzed. Figure 12 shows the error analysis of the second set of experiments. The predicted cutting force obtained by proposed model not only reduce the error of average cutting force, but also can be applied to the real-time cutting force of each tooth. The error of cutting force per tooth decreases from 20–30% to about 10%, indicating that the predicted cutting force of the proposed model is close to the measured cutting force. Although the error comes from the randomness of tool acceleration and measurement data, the prediction error is acceptable in data conversion.

In order to further verify the data conversion method, the stiffness parameters of cutter are changed by changing the diameter of cutter, whose stiffness parameters are shown in Table 7. Cutter B is used. The experimental conditions are shown in Table 8. The acceleration signal collected in test 1 is shown in Figure 13.

Then, the predicted cutting force are calculated. Result of cutting force comparison from test 1 is shown in Figure 14. The error comparison of all tests is shown in Figure 15.

Through the comparative analysis of cutting force and error, it is found that except repeating the occurred phenomena in previous section, here are few new anomalies in this section. The predicted cutting force obtained by the proposed method has high accuracy.

## 5. Conclusions

In this paper, a cutting force transition model considering the influence of the machine tool system is proposed, which can realize the transition from standard cutting data to actual machining data. Comparative experiments show that the accuracy of data transition by the proposed model is improved from about 30% to about 10%, which verifies the effectiveness of the method.

The practical significance of proposed method lies in the following. (1) The current cutting performance test data are obtained under specific cutting conditions, and the obtained cutting data have no standard to be based on and cannot be effectively applied. In this paper, the standard cutting data are obtained with orthogonal cutting by a special standard test table, which can be used as the basis for data transition. (2) There is no mapping from basic data to milling cutting data, which means the basic cutting data cannot be applied to actual parts production. In this paper, the influence of the tool system is considered in the data transition, and the effective data transition is realized.

For potential future research, firstly, we will further study the data transition method between heterogeneous process system and transfer the existing large amount of reliable data to the equipment with little information; secondly, we will improve the standard test table, and embed sensors to obtain more cutting process parameters, so as to obtain more complete information reflection. Moreover, better analysis and optimization of the process can improve the processing efficiency.

## Figures and Tables

**Figure 1 sensors-21-01340-f001:**
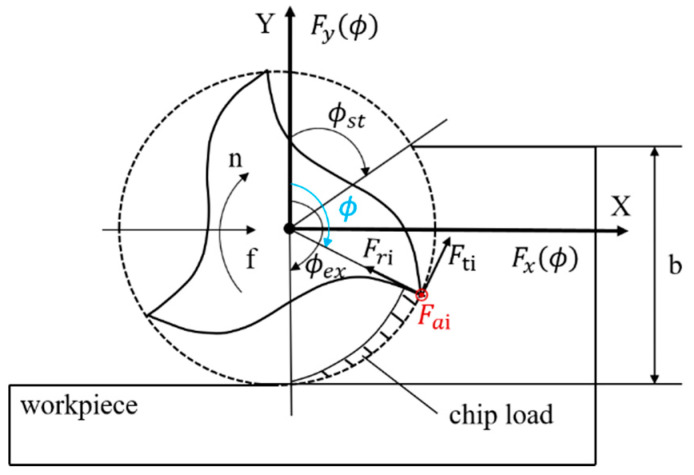
Geometry of milling process.

**Figure 2 sensors-21-01340-f002:**
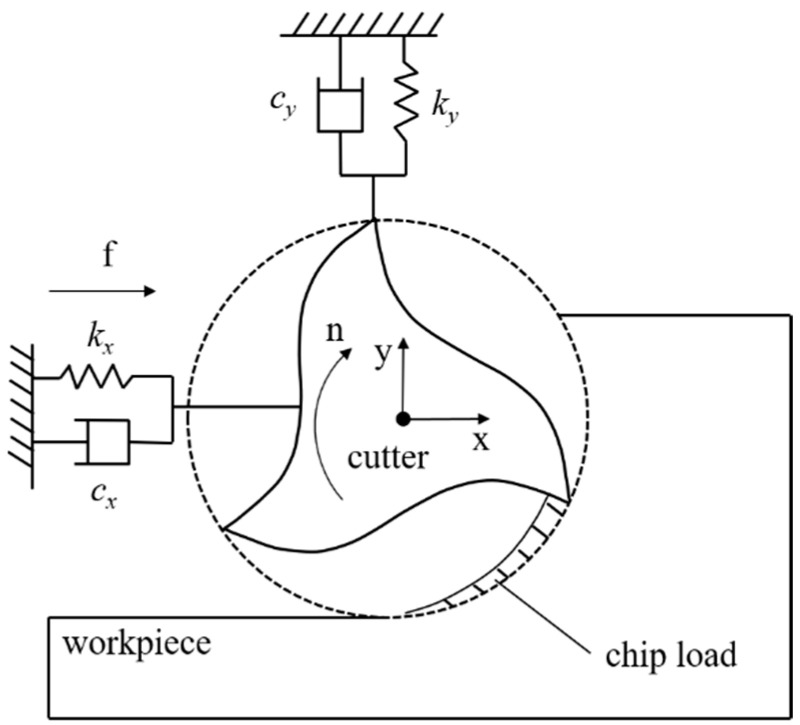
Milling cutter with two-degree freedom.

**Figure 3 sensors-21-01340-f003:**
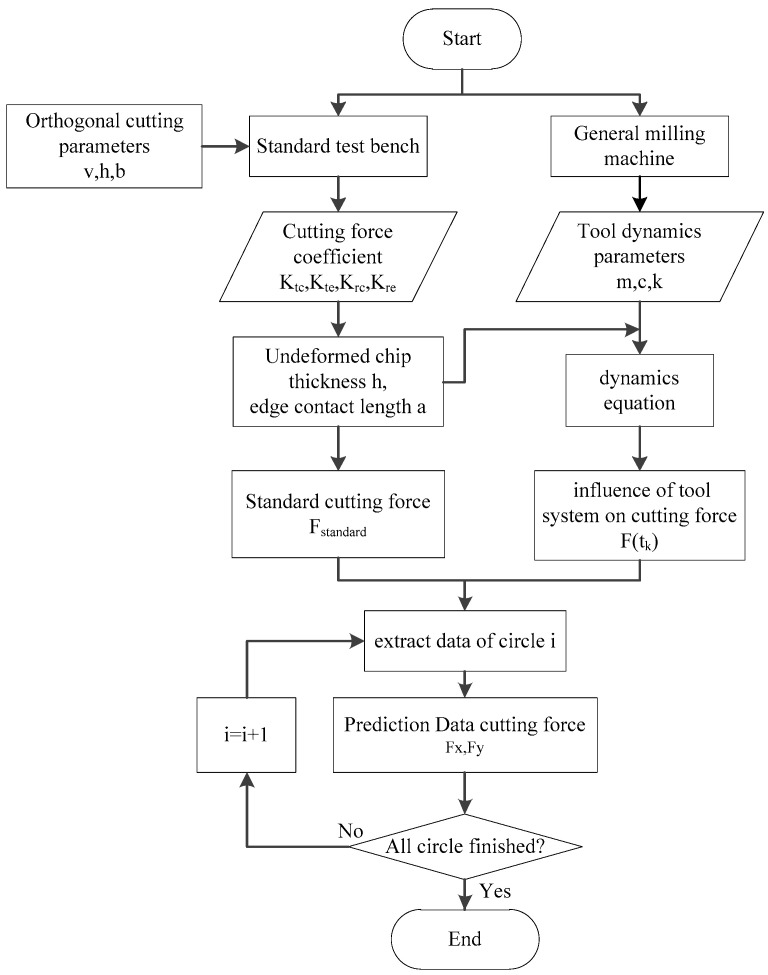
Flow chart of algorithm.

**Figure 4 sensors-21-01340-f004:**
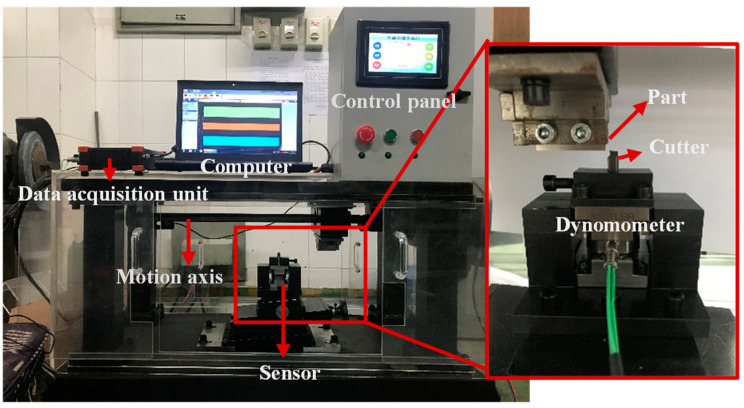
Setup of standard test table.

**Figure 5 sensors-21-01340-f005:**
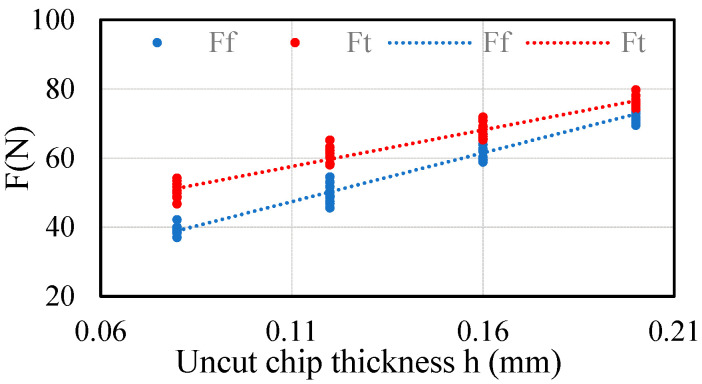
Cutting force of standard test table.

**Figure 6 sensors-21-01340-f006:**
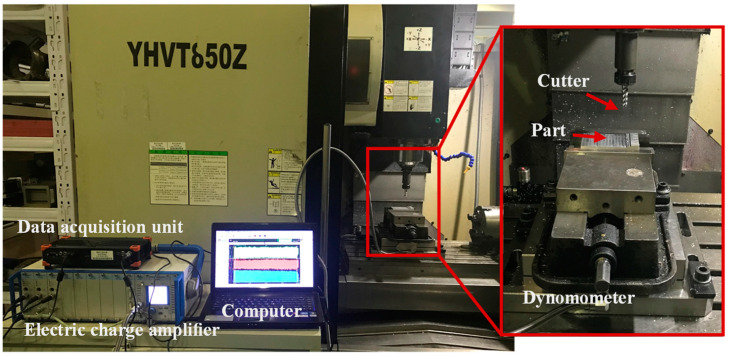
Setup of experiments.

**Figure 7 sensors-21-01340-f007:**
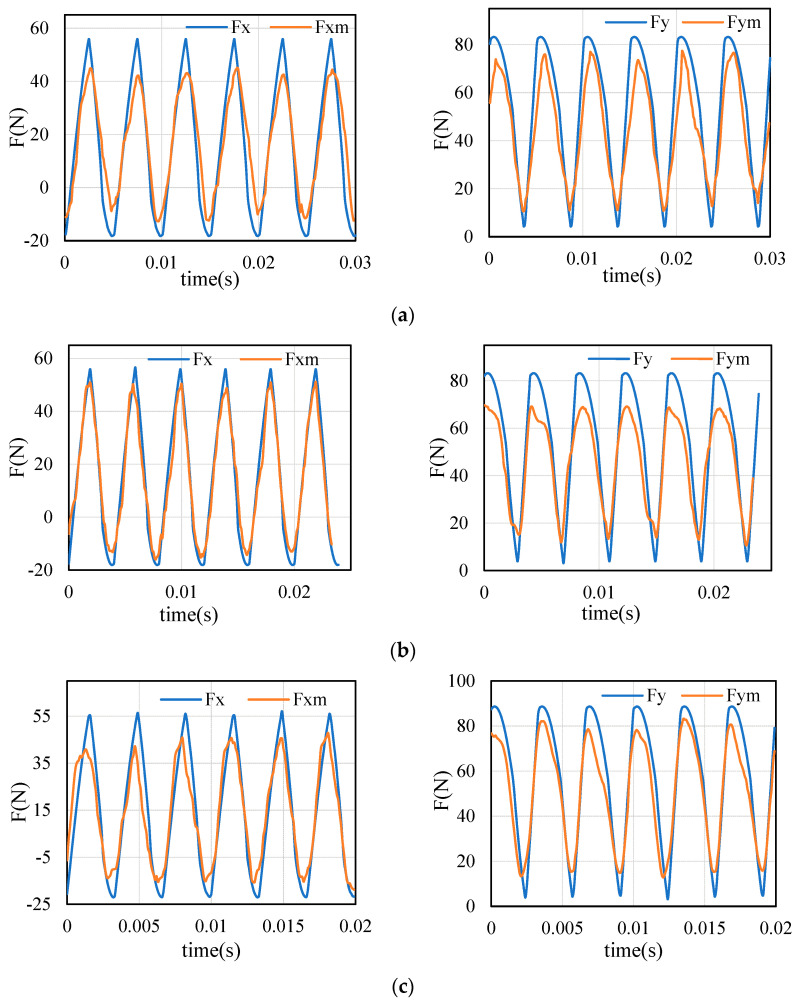
Comparison of measured cutting force and standard cutting force: (**a**) measured vs. standard (Test 2); (**b**) measured vs. standard (Test 6); (**c**) measured vs. standard (Test 12).

**Figure 8 sensors-21-01340-f008:**
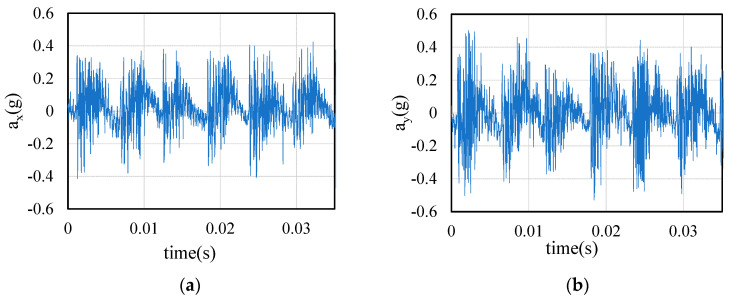
Acceleration signals from Test 2: (**a**) X direction; (**b**) Y direction.

**Figure 9 sensors-21-01340-f009:**
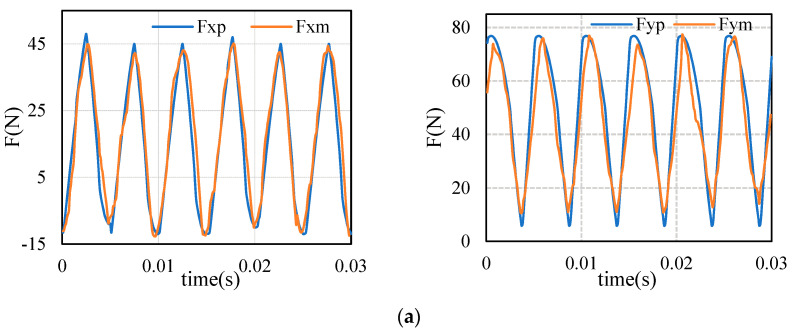
Comparison of measured cutting force and predicted cutting force: (**a**) measured vs. predicted (Test 2); (**b**) measured vs. predicted (Test 6); (**c**) measured vs. predicted (Test 12).

**Figure 10 sensors-21-01340-f010:**
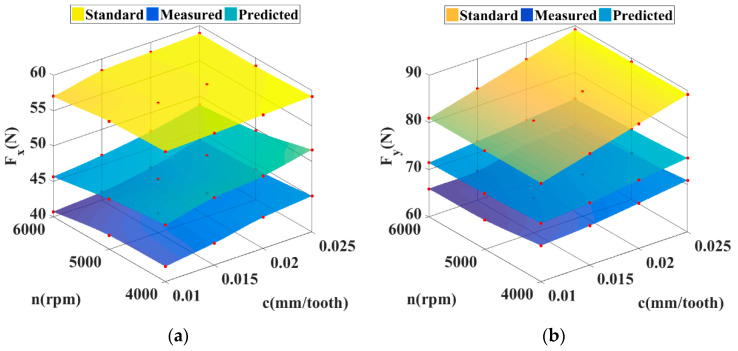
Comparison of cutting forces under different cutting parameters: (**a**) X direction; (**b**) Y direction.

**Figure 11 sensors-21-01340-f011:**
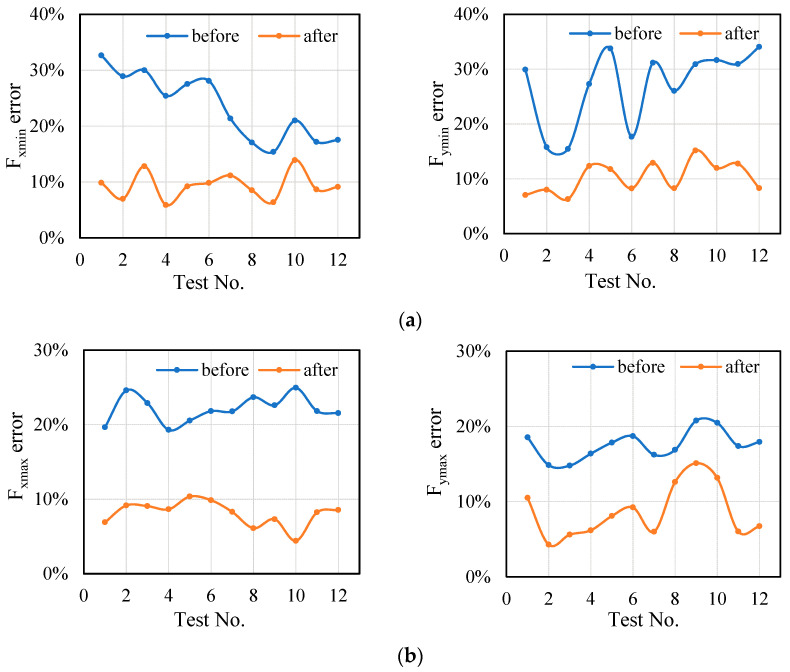
Data transition error before and after considering the influence of tool system: (**a**) Cutting force transition error in X direction; (**b**) Cutting force transition error in Y direction.

**Figure 12 sensors-21-01340-f012:**
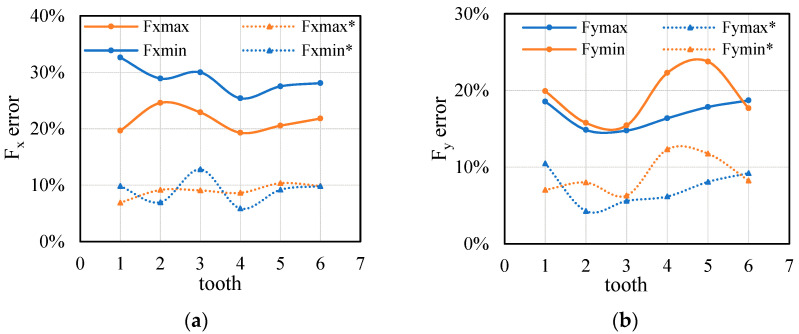
Data transition error comparison of each tooth cutter from Test 2: (**a**) Comparison of X direction error; (**b**) Comparison of Y direction error.

**Figure 13 sensors-21-01340-f013:**
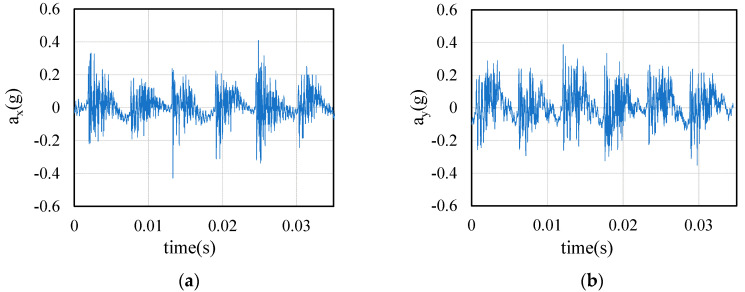
Acceleration signals from Test 1: (**a**) X direction; (**b**) Y direction.

**Figure 14 sensors-21-01340-f014:**
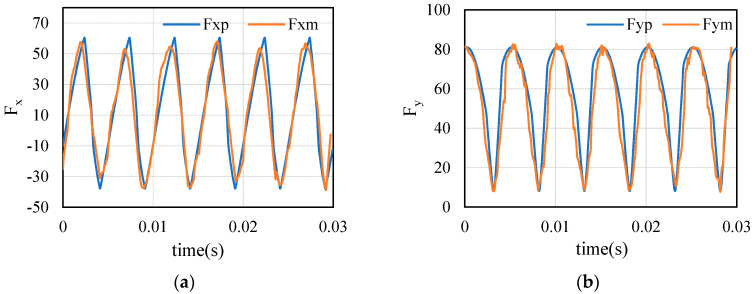
Comparison of measured cutting force and predicted cutting force in test 1.

**Figure 15 sensors-21-01340-f015:**
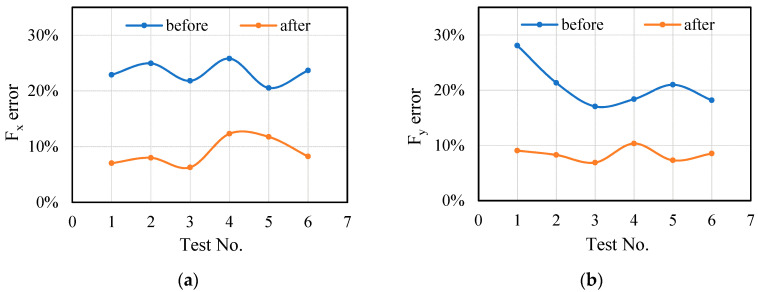
Data transition error before and after considering the influence of tool system: (**a**) Cutting force transition error in X direction; (**b**) Cutting force transition error in Y direction.

**Table 1 sensors-21-01340-t001:** Orthogonal cutting parameters.

Test No.	*v* (m/min)	*h* (mm)	*b* (mm)
1	40	0.08	1
2	40	0.12	1
3	40	0.16	1
4	40	0.20	1

**Table 2 sensors-21-01340-t002:** Identified cutting force coefficients.

$K_{tc}\;(N/{\mathrm{mm}}^{2})$	$K_{te}\;(N/\mathrm{mm})$	$K_{fc}\;(N/{\mathrm{mm}}^{2})$	$K_{fe}\;(N/\mathrm{mm})$
211.4	34.32	281.78	16.417

**Table 3 sensors-21-01340-t003:** Cutter geometry parameters.

Cutter	Helical Angle	Diameter (mm)	Length of Cutter (mm)	Length of Cutting Section (mm)	Number of Teeth
A	45°	12	75	36	3
B	45°	10	75	30	3

**Table 4 sensors-21-01340-t004:** Cutting parameters for milling process.

Test No.	*a_p_* (mm)	*a_e_* (mm)	*n* (rpm)	*f* (mm/min)
1	2	6	4000	120
2	2	6	4000	180
3	2	6	4000	240
4	2	6	4000	300
5	2	6	5000	150
6	2	6	5000	225
7	2	6	5000	300
8	2	6	5000	375
9	2	6	6000	180
10	2	6	6000	270
11	2	6	6000	360
12	2	6	6000	450

**Table 5 sensors-21-01340-t005:** Stiffness parameters of test tool (cutter A).

$m_{x}/m_{y}\;(\mathrm{kg})$	$c_{x}/c_{y}\;(N/(m/s))$	$k_{x}/k_{y}\;(N/m)$
2.6730 × 10^7^	235.8167	0.6891

**Table 6 sensors-21-01340-t006:** Error value of data transition before and after considering the influence of tool system.

Test No.		Before	After		Before	After
1	F_xmax_	19.65%	6.90%	F_ymax_	18.54%	10.48%
	F_xmin_	32.63%	9.84%	F_ymin_	29.91%	7.03%
2	F_xmax_	24.59%	9.15%	F_ymax_	14.85%	4.28%
	F_xmin_	28.91%	6.97%	F_ymin_	15.77%	8.00%
3	F_xmax_	22.89%	9.07%	F_ymax_	14.77%	5.60%
	F_xmin_	29.99%	12.81%	F_ymin_	15.44%	6.29%
4	F_xmax_	19.29%	8.64%	F_ymax_	16.37%	6.17%
	F_xmin_	25.40%	5.87%	F_ymin_	27.29%	12.33%
5	F_xmax_	20.54%	10.35%	F_ymax_	17.84%	8.08%
	F_xmin_	27.51%	9.19%	F_ymin_	33.76%	11.76%
6	F_xmax_	21.81%	9.86%	F_ymax_	18.70%	9.22%
	F_xmin_	28.08%	9.82%	F_ymin_	17.67%	8.24%
7	F_xmax_	21.77%	8.29%	F_ymax_	16.22%	5.99%
	F_xmin_	21.34%	11.15%	F_ymin_	31.15%	12.91%
8	F_xmax_	23.67%	6.11%	F_ymax_	16.87%	12.60%
	F_xmin_	17.06%	8.51%	F_ymin_	26.02%	8.26%
9	F_xmax_	22.60%	7.31%	F_ymax_	20.76%	15.10%
	F_xmin_	15.38%	6.36%	F_ymin_	30.89%	15.16%
10	F_xmax_	24.94%	4.41%	F_ymax_	20.46%	13.15%
	F_xmin_	20.99%	13.90%	F_ymin_	31.63%	11.96%
11	F_xmax_	21.81%	8.24%	F_ymax_	17.37%	6.02%
	F_xmin_	17.17%	8.65%	F_ymin_	30.93%	12.77%
12	F_xmax_	21.54%	8.55%	F_ymax_	17.92%	6.72%
	F_xmin_	17.52%	9.11%	F_ymin_	34.05%	8.29%

**Table 7 sensors-21-01340-t007:** Stiffness parameters of cutter B.

$m_{x}/m_{y}\;(\mathrm{kg})$	$c_{x}/c_{y}\;(N/(m/s))$	$k_{x}/k_{y}\;(N/m)$
1.5374 × 10^7^	235.8167	0.6891

**Table 8 sensors-21-01340-t008:** Cutting conditions for verification tests.

Test No.	*a_p_* (mm)	*a_e_* (mm)	*n* (rpm)	*f* (mm/min)
1	2	6	4000	120
2	2	6	4000	240
3	2	6	5000	150
4	2	6	5000	300
5	2	6	6000	180
6	2	6	6000	360

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
