# Peer review of "Cutting Force Transition Model Considering the Influence of Tool System by Using Standard Test Table"

_sensors, 2021, doi:10.3390/s21041340_

Round 1

Reviewer 1 Report

the paper have been improved from previous versions and English language reads well. The authors should check the following before the paper can be accepted:

  • Line 91-96 remove this paragraph as it doesn’t add any value to the paper.
  • There is a word ourselves in the paper, please change to, the table was developed or something similar.

Reviewer 2 Report

Dear Authors 

The reviewed article is very interesting. Below are some informative questions. 

1. Are the drawings (Fig. 1 and Fig. 2) presented in the article taken from other articles or authors? 

2. What measure of scattering did the Authors used in Fig. 5? 

Sincerely

Reviewer 3 Report

In this manuscript " Cutting force transition model considering the influence of tool system by using standard test table" by Xi Chen et al., the authors propose a cutting force conversion model that is combined with the tool dynamic properties and the standard cutting force to predict the cutting force in the actual milling process, which is an interesting work. However, there are still some problems to be solved.

  1. The English can be polished, if possible.
  2. The author collects the standard cutting force data by the orthogonal cutting experiment using the standard test table. How to define the “standard”? Is it suitable for all milling processes?
  3. Does the standard test table vary with the actual milling process? If so, what is the difference between the complexity of the proposed cutting force prediction method and that of the actual cutting force data collection?
  4. What are the meanings of the cutting force coefficients in Table 2, and how to obtain them?

Round 2

Reviewer 3 Report

the authors have addressed all my previous concerns well. I recommend its publicaiton on this journal

This manuscript is a resubmission of an earlier submission. The following is a list of the peer review reports and author responses from that submission.

Round 1

Reviewer 1 Report

Dear Authors, the presented material is very interesting, but requires improvement.

Figures 1 and 2 are elementary diagrams, they should be deleted.

Figures 5 and 6 are illegible and do not add anything new to the article, I suggest removing them.

In subsequent articles, the focus should be on citing new articles (not older than 10 years).

Reviewer 2 Report

First of all I woul like to mention that the manuscript reflects the efforts of a research group what is always good.

The writing needs a review, the speeling is correct most of the times, but many times the phrases don't make sense or are confuse. So I suggest that a native speaker do a review in this sense.

Table 2 are missing units.

For the purposes of the manuscript, the aluminum alloy must be better described as well as the cutting tool. At least for me SGO S550 has no meaning.

In my opinion, it was not clear the novelty of the work. What is the differece between a standard test table and a machine tool used to get the constants? There are no evidences that the methodology is appliable to other machine tool and cutting conditions. In this sense I believe that the work needs more test in different conditions, and even different materials. 

Reviewer 3 Report

The current paper investigates developing a cutting force transition model in milling process by taking into account the contribution of tool system dynamics. The authors develop the model using data acquisition and sensors to capture cutting forces/signals.

The topic is interesting and the experiments are conducted in good manner however the presenting style of the manuscript is unacceptable at its current state. The authors must revise the whole manuscript and rearrange the sections properly to make it easier to track and read. Also consider justifying the trends and results observed from the tests.

Also see extra suggestions below:

Line 21 “Machining is an important machining…” please revise this sentence and line.

Line 25 using comma instead of a dot when you say therefore

Line 27 I believe the authors mean cutting forces and not just a single force especially in milling process?

The manuscript does not read well, and it must undergo extensive English and grammar spelling checks. Many typos and mistakes are noticed in each line.

Lines 34-35 authors must give proper credit to the cited paper and explain how these different models based on tool vibration, workpiece or track worked instead of just mentioning them.

Line 38 what are these factors?

Line 39 first time an abbreviation appears it must be fully stated what it stands for. (NC: Numerical Machine).

There are so many issues in wording, check line 70, 111, 169, 178, 196, 272, 273

Line 201 why you call it Kistler sensor…this is a dynamoemter device which is well know for measuring forces and moments.

What is the model number of the dynamometer used in this study?

For table 1, why the authors specifically choose those cutting parameters for their study?

How many tests in total were carried out, how many tools were used and did the authors use a fresh tool with each new set of cutting conditions?

Please specify the exact names of the forces collected? They are called tangential/shear force or normal force and so on..

Ok now the dynamometer model is mentioned later. All these details need to be moved and grouped together. Adding them all over the manuscript is confusing and make it difficult to track.

Line 232-233 “which shows the results are fit with the calculated values, but the values are different.” Please explain more, this sentence is not clear.

Line 243-244 the authors must support this claim/justification and support using references.

Round 2

Reviewer 3 Report

The paper English still does not read well, without proper English editing and spelling check this paper can not be accepted. For example, see:

 Line 31-32 “put forward a very important and widely used model in cutting force modelling” just simplify the writing and be to the point. Adding unnecessary words makes it read poorly.

Line 98 “to consider the system influence”

Line 99 word mainly is not needed

Line 100 and many more, please check for English spelling and writing

Line 39-40 tell us what ref [14] and [15] did in their work and what they found not just say what was the main title of their work. Try to go in more depth and explain different models proposed and how they differ from each other.

Line 44-45 what do you mean by resulting in data that cannot be applied.

Line 59 font type changes in the word Considered

Line 226 again Kistler 9257B is a Multi-Component Dynamometer (write them this way to be clear)

Kistler 9347C is a 3 Axis load cell is a triaxial force transducer (write them this way to be clear)

Line 226 “aluminum alloy 7450” is this correct? I am not sure if this alloy grade is correct or if it have applications in aerospace.

Line 232 its not covered, its coated with TiAlN coating…

Line 261-262 “this is because workpiece is aluminium”…please explain more about this justification

Line 285 “as much as possible” is not scientific way to describe things

Line 300 “close to each other” they are not close to each other, perhaps you meant to say they are similar in values

Overall I think the work done here is interesting and can be published however, the authors need to submit a fresh copy with extensive English editing and spell checking. Also, it is strongly recommended to keep the explanation of findings and results short and to the point, using scientific wording and support any claims by references.

Also compare your findings with past work if possible.

Round 3

Reviewer 3 Report

the authors have answered all the questions, however it was difficul to track all changes with the track changes left on.

1) Al7575 is not the most widely used alloy in aerospace, unless authors can support this with a reference then this should be removed. 

2) The tool use is SGO S550, what does this have to do with the effect on results outcome?

3) Line 311-312 is not good justification for the outcome of the finding.

Writing style is still an issue and must be improved.
